# Early Effects of Low Molecular Weight Heparin Therapy with Soft-Mist Inhaler for COVID-19-Induced Hypoxemia: A Phase IIb Trial

**DOI:** 10.3390/pharmaceutics13111768

**Published:** 2021-10-22

**Authors:** Mustafa Erelel, Mert Kaskal, Ozlem Akbal-Dagistan, Halim Issever, Ahmet Serhan Dagistanli, Hilal Balkanci, Merve Sinem Oguz, Aygun Qarayeva, Meltem Culha, Aybige Erturk, Nur Sena Basarir, Gokben Sahin, Ali Yagiz Uresin, Ahmet Ogul Araman, Alpay Medetalibeyoglu, Tufan Tukek, Mustafa Oral Oncul, Ayca Yildiz-Pekoz

**Affiliations:** 1Department of Chest Diseases, Faculty of Medicine, Istanbul University, 34104 Istanbul, Turkey; merelel@istanbul.edu.tr (M.E.); ahmet.dagistanli@istanbul.edu.tr (A.S.D.); hilal.balkanci@istanbul.edu.tr (H.B.); sinemoguz@istanbul.edu.tr (M.S.O.); aygun.garayeva@istanbul.edu.tr (A.Q.); 2Department of Pharmacology, Faculty of Medicine, Marmara University, 34854 Istanbul, Turkey; mert.kaskal@marmara.edu.tr; 3Department of Pharmaceutical Technology, Faculty of Pharmacy, Istanbul University, 34116 Istanbul, Turkey; ozlemakbal@istanbul.edu.tr (O.A.-D.); meltemculha@ogr.iu.edu.tr (M.C.); aybige.erturk@istinye.edu.tr (A.E.); nursena.basarir@ogr.iu.edu.tr (N.S.B.); gokbensahin@trakya.edu.tr (G.S.); aramana@istanbul.edu.tr (A.O.A.); 4Department of Public Health, Faculty of Medicine, Istanbul University, 34104 Istanbul, Turkey; hissever@istanbul.edu.tr; 5Department of Pharmaceutical Technology, Faculty of Pharmacy, Istinye University, 34010 Istanbul, Turkey; 6Department of Pharmaceutical Technology, Faculty of Pharmacy, Trakya University, 22130 Edirne, Turkey; 7Department of Medical Pharmacology, Faculty of Medicine, Istanbul University, 34104 Istanbul, Turkey; yagiz@istanbul.edu.tr; 8Department of Internal Diseases, Faculty of Medicine, Istanbul University, 34104 Istanbul, Turkey; alpay.m@istanbul.edu.tr (A.M.); tufan.tukek@istanbul.edu.tr (T.T.); 9Department of Infectious Diseases and Clinical Microbiology, Faculty of Medicine, Istanbul University, 34104 Istanbul, Turkey; oraloncul@istanbul.edu.tr

**Keywords:** COVID-19, low-molecular-weight heparin, soft-mist inhaler, pulmonary, anti-coagulant

## Abstract

In COVID-19-induced acute respiratory distress syndrome, the lungs are incapable of filling with sufficient air, leading to hypoxemia that results in high mortality among hospitalized patients. In clinical trials, low-molecular-weight heparin was administered via a specially designed soft-mist inhaler device in an investigator initiated, single-center, open-label, phase-IIb clinical trial. Patients with evidently worse clinical presentations were classed as the “Device Group”; 40 patients were given low-molecular-weight heparin via a soft mist inhaler at a dose of 4000 IU per administration, twice a day. The Control Group, also made up of 40 patients, received the standard therapy. The predetermined severity of hypoxemia and the peripheral oxygen saturation of patients were measured on the 1st and 10th days of treatment. The improvement was particularly striking in cases of severe hypoxemia. In the 10-day treatment, low-molecular-weight heparin was shown to significantly improve breathing capability when delivered via a soft-mist inhaler.

## 1. Introduction

The clinical severity of COVID-19 varies from mild (~80%) to life-threatening pneumonia [1]. Almost 20% of COVID-19 patients suffer from hypoxemia, the predominant cause of hospitalization and mortality [2,3]. Up to 24% of hospitalized patients require invasive mechanical ventilation due to lung injury resulting from acute respiratory failure and hypoxemia [2]. Interventions to overcome hypoxemia include inhaled corticosteroids, pulmonary vasodilator therapies (i.e., nitric oxide and prostaglandins), anti-inflammatory medications (i.e., tocilizumab, anakinra, sarilumab, siltuximab etc.) and oxygen therapy [4,5].

The repurposing of available medicines is the fastest option to make new pharmacotherapies, including antivirals [2], heparin-derivatives [6] and corticosteroids [7].

Our clinical study utilizes the liquid form of low-molecular-weight heparin (LMWH), delivered via a soft mist inhaler (PulmoSpray^®^) to reduce side-effects and to achieve a higher accumulation in the targeted organ. LMWH has advantages over current treatments of hypoxemia, such as anticoagulant, anti-inflammatory, mucolytic and anti-viral properties, particularly in the treatment of SARS-CoV [8,9], but also Zika [10], Herpes Simplex [11], influenza [12] and HIV [13,14]. Mycroft-West et al. showed in vitro that the addition of heparin (100 μg/mL) to vero cells inhibited the internalization of SARS-CoV-2 by 70% [15]. We propose that the inhalation of heparin derivatives via a soft-mist inhaler is beneficial to cure hypoxemia-induced acute lung injury, a life-threatening complication of COVID-19. Although published studies support only the in vitro effect [16,17] and conventional usage [16,18] of unfractioned heparin (UFH) or LMWH on COVID-19 patients, recent clinical literature has focused primarily on the antiviral efficacy of heparin in COVID-19. Given that inhaled administration of heparin has been proven to be safe in humans, the hypothesis of the research topic may open up new methods of treatment [19].

Our research hypothesizes that, if the lungs are the primary target of the virus, therapies should focus on local treatment. Our research has two objectives. On the one hand, the efficacy and safety of the formulation must be certified. On the other hand, the vessel of delivery (i.e., the inhaler) must fulfil the safety and performance criteria. Compliance among non-ICU (Intensive Care Unit) patients, the spraying capacity of the inhaler as well as its safety and practicality in a clinical setting are parameters to consider. Under pandemic conditions, particularly in clinical settings, the risk of “cross contamination by the inhaler” must be eliminated.

Our research aims are as follows:i.To alleviate COVID-19-induced hypoxemia and improve patient respiratory capacity.ii.To reduce the death toll by hypoxemia to the lowest possible level.iii.To eliminate the side effects associated with the current protocols.iv.To reduce intubation rates.v.To improve clinical care capability and reduce cross-contamination risk to health personnel.vi.To achieve the highest drug concentration in lungs [20].

If this novel delivery method, coupled with the proposed novel formulation, proves effective, its potential is expected to not be limited to the treatment of SARS-CoV-2.

## 2. Materials and Methods

### 2.1. Material

PulmoSpray^®^ (Figure 1) for in vitro and clinical applications were kindly donated by Resyca BV (Enschede, The Netherlands). LMWH (Enoxaparin sodium; Oksapar 4000 anti-Xa IU/0.4 mL) was kindly donated by Kocak Farma AS (Istanbul, Turkey).

### 2.2. Clinical Study Design and Patients

This research was an investigator-initiated, single-center, open-label phase IIb trial conducted at the Istanbul Medical Faculty Hospital. Since the inhalation of UFH via a nebulizer was shown to be highly effective for acute lung injury and acute respiratory damage [21], patients with a relatively more severe clinical course were given priority, and were classed as the “Device Group”. The present study was approved by the Istanbul Medical Faculty and Turkish Medicines and Medical Devices Agency Clinical Research Ethics Committee (approval number E-66175679-514.03.01-328141, 27 January 2021). Also, this trial is publicly available at ‘ClinicalTrials’ under registration number NCT04990830. Patients in the Device Group were given LWMH via a soft mist inhaler at a dose of 4000 IU/0.4 mL per administration twice a day, plus the standard treatment. Each inhaled LMWH application was performed manually for approximately 2 min under the supervision of a health professional. Standard treatment was given to both groups. The Control Group received systemic treatment, i.e., only subcutaneous LMWH, the Device Group received LMWH inhalation in addition to systemic treatment. The administered inhaled LMWH is expected to accumulate locally, and therefore not to increase the systemic dose. A full list of criteria can be found in Table 1.

The sample size calculation for this clinical trial was performed by evaluating the necessary patient population for the study to be effectively carried out. It was determined that 40 patients would be included in the treatment group and 40 in the control group. The sample size for this study was based on a comparison of the Day 10 discharge rates between both groups. In this descriptive study, the sample size was determined completely hypothetically in order to guide the future phase III study. According to this hypothesis, since 85% of patients were discharged on day 10 following administration of inhaled LMWH, it was estimated that the discharge rate at Day 10 would be 48% for patients who took only standard therapy. On this basis, it was decided that inclusion of at least 37 patients in each group would provide an adequate sample size with 90% power and 5% error. The sample size calculation between two groups was performed based upon O_2_ requirements; namely, 50% was the beginning point for more than one sample size calculation, and the largest sample size calculated was proposed as the sample size of the study. For each group, 37 was calculated as the minimal sample size when the difference was 35% between the two groups, and the study scope was conjectured based on 40 patients in each group. Since more than one possible O_2_ change between the two groups could take place, the sample size was hypothesized to be sufficient for as few as 32 patients in each group [22]. The assignment of participants was performed in accordance with the flow chart given in Figure 2.

Additional detail on the clinical study design is provided on Appendix A.

At the beginning of the trial, patients were categorized into five “severity” levels (Table 2) based upon their oxygen therapy requirements. Above 95% was the accepted oxygen saturation (SpO_2_) value to prevent or reverse organ damage and maintain the necessary oxygenation circulation; thus, this was determined as the primary end point [23]. If the oxygen saturation was higher than 95%, patients were defined as “Room Air”; this group consisted of patients with milder clinical presentation. At the end of the trial, it was found that 13 patients had changed to “Room Air” status in the Control Group. In the Device Group, 25 patients were upgraded to this status. The decrease in severity levels for the device group was significant (*p* < 0.01) compared to that of the Control Group.

Upon patient admission, low-dose computerized tomography was performed. Parenchymal data were categorized into severity degrees based on following criteria: lobe involvement, involved area of lobe, and patch or diffuse, as shown in Table 3.

### 2.3. Outcomes

The primary outcome was the assessment of oxygen saturation and hypoxemia status at the end of the 10-day treatment to evaluate the efficacy of our hypothesis. The need for oxygen supply indicates whether the proposed treatment had given rise to a considerable difference in comparison with the standard in terms of the number of patients taken out of intubation and intensive care.

On Day 1, the rationality for the oxygen supply method was established based on oxygen saturation, fever and biochemical clinical parameters, including CRP, ferritin, D-dimer, neutrophil count, lymphocyte count, and the neutrophil to lymphocyte ratio.

### 2.4. Mechanism and In Vitro Lung Deposition Studies of Pulmospray^®^

Pulmospray^®^ belongs to a recently classified group of inhaler devices described as soft mist inhalers that can deliver aerosolized solutions. The LMWH solution must be converted into droplets to achieve an inhalable aerosol of the required size for the drug solution to fit into the soft mist inhaler system. The soft mist mechanism of the inhaler creates a mechanical force by compressing a spring, thereby causing a piston to compress. The driving force is based on the principle of forcing the drug solution through a series of “very small micro-nozzles” to form an aerosol mist. Moreover, the energy required for aerosol generation comes from the inhaler itself, and critically for COVID-19 patients, is independent of the patient’s respiratory capacity [24].

The next generation impactor (NGI) is an internationally recognized cascade impactor device that is used for classifying aerosol particle into size fractions to test the performance of inhalers. In order to evaluate the aerosol performance of the LMWH solution, an NGI (Copley Scientific, Nottingham, UK) with a USP metal induction port was utilized. The NGI setup and procedures described in the Inhaler Testing Guide and EP. 2.9.18 (EP Monograph 2.9.18, 2010) were followed [25]. The NGI aerosol parameters evaluated for LMWH solutions were fine particle dose (FPD), fine particle fraction (FPF), median mass aerodynamic diameter (MMAD) and geometric standard deviation (GSD). A steady inhalation flow of 20 L/min (±5%) was applied through the NGI configuration for a duration of 3 s per puff with 1 mL sample at 5 °C (*n* = 6). The amounts deposited in each cup and remaining in the system were measured by the colorimetric method.

### 2.5. Data and Statistical Analysis

In vitro data were compared via one-way analysis of variance (ANOVA) (GraphPad Software, La Jolla, 8.1.1., San Diego, CA, USA). A clinical data analysis was performed using SPSS version 23.0 software (SPSS Inc., Chicago, IL, USA). The conformity of the measurements to normal distribution was established with Kolmogorov-Smirnov. Variables conforming to normal distribution were used for the *t*-test for independent groups, while nonconfirming variables were used for the Mann-Whitney U test. Pre/post-treatment variables were evaluated with the paired sample *t*-test and the Wilcoxon signed-rank test. A Chi-square analysis was applied to the categorical variables between the two groups. Statistical significance was *p* < 0.05, two-tailed.

## 3. Results

### 3.1. Clinical Study Results

The average radiological severity scores were 5.6 ± 1.5 for patients in the Device Group and 6.4 ± 1.8 for those in the Control Group, meaning there was no significant difference in radiological severity between the two groups.

Upon administration, basic patient characteristics were taken. These values are presented in Table 4.

The patients of both groups presented with primarily respiratory distress, which is typical for COVID-19: incessant coughing, sputum and shortness of breath, high fever (above 38 °C), and extreme fatigue (Table 5). Clinical parameters, i.e., peripheral oxygen saturation along with CRP, ferritin, leukocyte count, the neutrophil/lymphocyte ratio and other laboratory parameters are shown in Table 5.

A peripheral saturation value of 95% or above was set as “normal” for both groups, and any value below this as “hypoxemia”.

Of the laboratory parameters, CRP was significantly higher (<0.01) in the Control Group, while ferritin, leukocyte, and the neutrophil/lymphocyte ratio were significantly higher (<0.01) in the Device Group. The upper limits of the D-Dimer value did not differ significantly between two groups (Mann-Whitney U). The Device Group included more severe patients compared to the Control Group based on the aforementioned parameters.

Clinically, shortness of breath and sputum production were significantly higher in the Device Group (<0.01). Coughing was not significantly different between the two groups.

The mean time from onset of symptoms to hospitalization in the Device Group was 3.50 ± 1.99 (Coefficient of Variation: CV: 56.85%) and 4.40 ± 4.23 (Coefficient of Variation: CV: 96.13%) days in the Control Group. In terms of clinical symptom scoring, the Device Group had a significantly higher symptom score, meaning that (statistically on average) members of this group may be said to have experienced COVID-19 “more severely”. Since inhaled LMWH has been shown in previous studies to be effective in improving lung injury [26,27], patients with more severe symptoms were given priority (by medically informed ethical choice) to be placed in the Device Group, and thus to receive inhaled LMWH therapy (Table 5).

Patient hypoxemia and peripheral oxygen saturation values were measured on the 1st and 10th days and based on patient responsiveness to the supply method, a severity index was established, where each level implies a different method of oxygen supply, with 95% being the threshold.

By the end of the 10-day treatment, a marked difference existed between the Device and Control Groups in terms of the number of patients in the “room air” category. This difference provides a basis for comparison between the device as hypothesized in this study and existing methods of oxygen supply.

Patients in the Device Group needed a highly significant (*p* < 0.01) intensive oxygen therapy to overcome hypoxemia. Improvement in patient hypoxemia by the 10th day, as evaluated by the method of oxygen supply, is shown in the Table 6. In the Device Group, 13/13 patients with hypoxemia who were supplied oxygen via nasal cannula were normoxemic by the end of the treatment. Of the Device Group, 16/35 cases (45.7%) had improved by one degree, 12/35 cases (34.3%) by two degrees and 3/35 cases (8.6%) by three degrees.

In the Control Group, however, the 10-day period showed a more heterogeneous outcome. For instance, in the nasal cannula subgroup, 4/15 cases (26.6%) showed no change in status, while three patients had to be intubated at some point within the 10-day period due to a deterioration in their condition. In terms of overall improvement rate, 14/40 cases (35%) improved by one severity level, 2/40 cases (5%) improved by two and only 1/40 case (2.5%) improved by three severity levels. In contrast, three patients in the Device Group improved by three severity levels. The greatest contrast was recorded in improvement by two levels: in the Control Group, only 5 % improved by two severity levels following standard therapy, whereas in the Device Group, 34.2 percent improved by two degrees. In particular, the fact that three cases (7.5%) with nasal cannula subsequently required intubation implies that the outcomes of current treatments may be quite heterogeneous in terms of patient response. Even if many individuals recover following standard treatment, some patients nonetheless deteriorate into “more severe” levels (i.e., requiring intubation).

The reduction in the oxygen supply amount in the Device Group was statistically significant compared with that of the Control Group. In the Device Group, there were no cases of intubation following treatment, whereas in the Control Group, three patients had to be intubated de novo, indicating that the probability of intubation risk was not predictably reduced for the Control Group. In terms of clinical respiratory symptoms on Day 1, the improvement performance of the Device Group was better than that of the Control Group (Table 6). The reduction in oxygen supply to correct hypoxemia in the Device Group was statistically significant compared with that of the Control Group (*p* < 0.01). In the subgroup analyses based on the oxygen supply method, the significance of the treatment was borderline in nasal cannula, whereas the so-called “improvement leap” (difference in improvement) was even more pronounced for more severe patients in the Device Group who received oxygen via a reservoir oxygen mask or high flow oxygen therapy (*p* < 0·01).

In summary, improvements in the Device Group were greater across all levels, meaning that patients benefitted more significantly from the use of the proposed device. Improvements in the Device patients were more homogenous and predictable, whereas those in the Control Group changes were more sporadic and unpredictable.

### 3.2. In Vitro Lung Deposition Study Results

The deposition of LMWH inhalation solution at each stage of NGI following inhalation via a soft mist inhaler is shown in Figure 3. The data was presented as the percentage of drug deposited in the universal induction port (UIP, throat) and at each stage of NGI over the administered dose. This dose was defined as the total amount of drug recovered from the throat and the stages of the NGI. After application, 57.08 ± 2.07% of the droplets were concentrated over 3–5 stages of the impactor, of which the corresponding cutoff diameter was 4.76–1.74 μm (Figure 3). The calculated FPF (fine particle fraction) was 44.4% ± 2.3 μm, MMAD (mass median aerodynamic diameter) was 5.37 ± 0.11 μm, and GSD (geometric standard deviation) was 1.63 ± 0.03 μm. It has been reported that a MMAD within a range of 1–5 μm can facilitate the retention of drugs in the lower respiratory region [28]. These results indicate that most of the LMWH inhalation solution formed into droplets that could be deposited in the bronchus and the bronchiole region of the lung.

## 4. Discussion

The current study aims to treat hypoxemia resulting from COVID19-induced acute respiratory distress syndrome (ARDS) by inhalation of LMWH via a soft-mist inhaler. This method is expected to provide a viable alternative to systemic treatments. By the 10th-day of treatment, a tangible reduction in hypoxemia was observed in the Device Group. The present research comprises a pilot study intended to show the efficacy of inhaled-LMWH on COVID-19-induced lung injury. Considering these positive results, this application route of LMWH should be further evaluated with a larger number of patients in a multicenter trial. To our knowledge, this study is the first clinical trial showing that inhaled LMWH improves hypoxemia in COVID-19 patients.

The hypothesis is that COVID-19-induced hypoxemia is treatable by locally targeting ARDS via drug inhalation [29,30]. Generally, ARDS-induced hospital mortality is estimated to be within the range of 35–40% [31]. In case of COVID-19-induced ARDS, it is crucial to note that this value increases up to 66% [32], and therefore, that this factor constitutes the highest mortality risk among COVID-19 complications.

The oxygen requirement of patients in the Device Group on Day 1 was higher than that in the Control Group. It was a deliberate decision, taken by the researchers, to place more severe patients in the Device Group, because it was stipulated that the proposed treatment could offer a last resort for otherwise critical patients. This argument presupposed that the patients have a right to receive innovative therapies under extraordinary pandemic conditions. It was also argued that inhaled LMWH could have positive effects on the prognosis of these patients, where standard protocols would not suffice. The results confirmed our expectations, i.e., 25 (75.8%) out of 33 patients in the Device Group who needed complete oxygen support at the start of the study were able to breathe in “room air” without oxygen support on day 10. In the Control Group, however, only 13 (54.1%) out of 24 patients who needed oxygen support at the beginning were upgraded to the “room air” category. This comparison provides a clear picture of the positive effect of inhaled-LMWH on oxygen demand. It is also noteworthy that the clinical status of the patients in the Device Group significantly improved [33].

Among devices which are available on the market, heparin can be applied in liquid dosage form or can be inhaled, either via a nebulizer or a soft-mist inhaler. Soft-mist inhaler technology was preferred in order to achieve the highest possible drug accumulation in the target area and to provide an “enclosed system”, whereby environmental contamination, caused by saliva dispersion in the air, is minimized [34]. The risk of environmental contamination in a clinical setting, particularly for healthcare professionals, was therefore drastically reduced. The soft-mist inhaler utilized in this trial is disposable; therefore, it is highly practical for pandemic conditions. Its mechanics allow the dose to be adjusted at each application. Thus, medical staff could adjust the dosage in response to a given patient’s changing status and individual needs. This soft-mist inhaler was redesigned by our research group to achieve a rate of retention in the lungs which was at least double that of the nebulizer devices [35].

Although there are very few published studies on inhaled heparin as a therapeutic agent for the treatment of COVID-19, several clinical studies applying nebulizers are currently being conducted [36]. Our research differs fundamentally by virtue of the following objectives:

(1)This study focused on LMWH to eliminate the need for mechanical ventilation by pro-actively curing hypoxemia itself.(2)The specifically designed device of delivery met the mechanical requirements for targeted delivery with the highest retention rate.

We believe that there are concomitant mechanisms initiated by heparin which positively act on the lungs of COVID-19 patients. Anticoagulant activity of heparin in COVID-19 patients was recently documented [37]. Moreover, a recent paper presented the case for the use of heparin, citing its beneficial properties, namely anti-inflammatory, anti-viral, histone neutralizing and heparanase (HPSE) inhibiting properties, in addition to its anticoagulant effect [38].

We speculate that the anticoagulant activity of heparin, as well as the four other properties mentioned above and discussed in more detail below, may have contributed, in conjunction with the standard therapies, to the positive outcomes of our trial by subduing COVID-19 complications.

In nonclinical studies, heparin/LMWH was shown to be highly effective in the treatment of COVID-19 by various mechanisms, including but not limited to, the anticoagulant and anti-inflammatory activity of the substance. The antiviral activity of heparin/LMWH has been shown to reduce viral entry into host cells [17,39].

Additionally, recent publications have proposed that heparin has an inhibitory effect on HPSE activity. The significance of this inhibition lies within the ability of HPSE to penetrate or affect the endothelial barrier, in which case vascular leakage of fluids and proteins would take place [40,41]. Moreover, if histones are present in the extracellular space when cell death occurs, they will induce inflammatory expression that is highly cytotoxic. Since histones, which are positively-charged proteins, are conserved, negatively-charged heparin is expected to neutralize this cytotoxic effect, and thus, to reduce potential organ damage [42,43]

This study has several limitations. Firstly, the Device and Control Groups were deliberately nonrandomized due to severe pandemic conditions. Secondly, changes in the biochemical parameters after the 10-day trial meant that CRP, ferritin, D-Dimer, and lymphocyte counts were not considered. The rationale of this exclusion was that from a statistical perspective, the 10-day period was too short to quantitatively account for parametric changes. Nonetheless, within this limitation, the observed biochemical parametric changes confirmed an overall improvement in patient clinical respiratory condition. In contrast, changes in the oxygen levels could be quantified with statistical analyses.

In conclusion, this study proposes that inhaled LMWH via a soft-mist inhaler significantly improves hypoxemia in COVID-19 patients. Soft-mist inhaled LMWH was well-tolerated and markedly decreased the need for the oxygen treatment (compared to reservoir masks, high-flow oxygen therapy) at the end of the 10-day treatment. This study group is presently carrying out a follow-up trial with a larger patient group to establish the extent to which soft-mist LMWH attenuates lung injury and hypoxemia in COVID-19.

## 5. Patents

This work has a pending Turkish patent (TR-2020/12816 and TR-2021/00552) and is subject to the Patent Cooperation Treaty (PCT) (PCT/TR2021/050630).

## Figures and Tables

**Figure 1 pharmaceutics-13-01768-f001:**
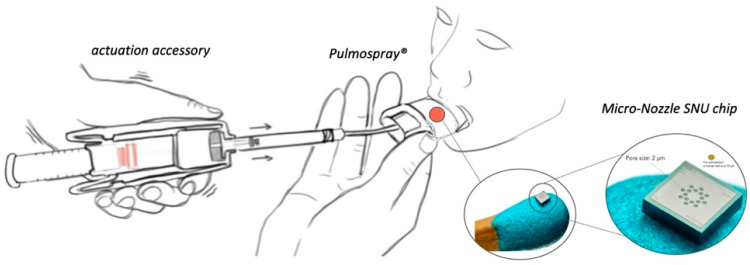
Illustration of Pulmospray^®^.

**Figure 2 pharmaceutics-13-01768-f002:**
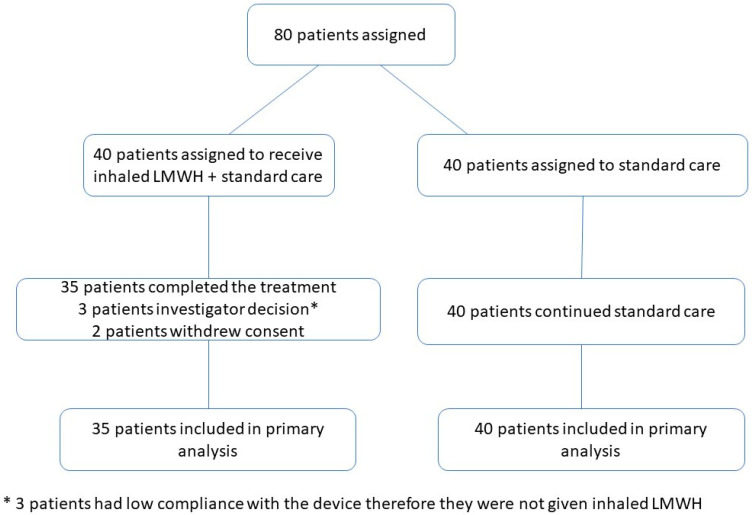
Participant Flow Chart.

**Figure 3 pharmaceutics-13-01768-f003:**
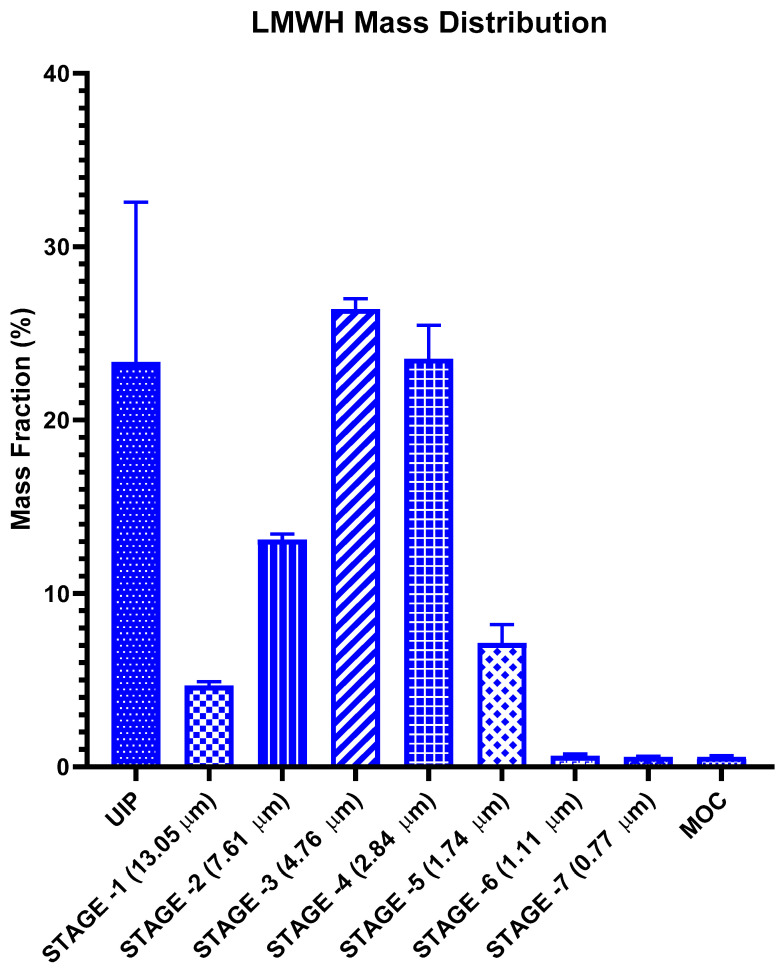
Lung deposition of LMWH solution in each of the NGI stages (*n* = 6, *p* < 0.01).

**Table 1 pharmaceutics-13-01768-t001:** Inclusion and Exclusion Criteria for All Patients.

**Inclusion Criteria**	**Written Informed Consent**
	Positive RT-PCR ^1^ test of nasopharyngeal swab for COVID- 19, and pneumonia confirmed by a CT ^2^.
	Negative RT-PCR test of nasopharyngeal swab for COVID- 19, but radiological and biochemical examinations unambiguously suggest COVID-19, when other possible diagnoses have been excluded.
**Exclusion Criteria**	Patients who are pregnant
	History of heparin and associated drug allergies

^1^ RT-PCR: Reserve Transcriptase-Polymerase Chain Reaction; ^2^ CT: Computerized Tomography.

**Table 2 pharmaceutics-13-01768-t002:** Severity Levels of Patients Based on the Mode and the Quantity of Oxygen Supplied.

Severity Level	Definition
0: Room Air	If the patient can breathe comfortably in room air.
1: Nasal Cannula	If peripheral oxygen saturation improves with an oxygen therapy up to 6 L/min via nasal cannula.
2: Reservoir Oxygen Mask	If condition can be improved with a 500 mL reservoir oxygen mask with 15 L/min oxygen treatment.
3: High Flow Oxygen	If condition can be improved with high flow oxygen therapy.
4: Intubation	If the intubation is the only choice.

**Table 3 pharmaceutics-13-01768-t003:** Radiological Severity Index.

Degree of Severity	Definition
1	One lobe less than 25% of lobe area
2	One lobe more than 25% of lobe area
3	Unilateral and less than one lobe less than 25% of each lobe area
4	Unilateral and more than one lobe less than 25% of each lobe area
5	Bilateral patch lesions on all lobes
6	Bilateral, all of one but not all lobes
7	Bilateral, all lobes, diffuse but less than 25% of each lobe area
8	Bilateral, all lobes, diffuse and 25–50% of each lobe area
9	Bilateral, all lobes, diffuse and 50–75% of each lobe area
10	Bilateral, all lobes, diffuse and more than 75% of each lobe area

**Table 4 pharmaceutics-13-01768-t004:** Baseline Patient Characteristics.

Characteristics	Device Group	Control Group
Demographics		
Age (y)	60.02 ± 10.04	59.62 ± 14.60
	*n* = 35	*n* = 40
Female	*n* = 15 (43.0%)	*n* = 15 (37.5%)
Male	*n* = 20 (57.0%)	*n* = 25 (62.5%)
Body Mass Index (kg/m^2^)	29.3 ± 4.5	30.4 ± 5.1
	*n* = 35	*n*= 40
Co-Morbidities		
Tobacco Smoking	*n* = 3 (8.5%)	*n* = 6 (15.0%)
COPD ^1^	*n* = 3 (8.5%)	*n* = 2 (5.0%)
Cardiac Disease	*n* = 7 (20.0%)	*n* = 9 (22.5%)
Diabetes Mellitus	*n* = 8 (22.8%)	*n* = 10 (25.0%)
Hypertension	*n* = 11 (31.4%)	*n* = 18 (45.0%)

^1^ COPD: Chronic Obstructive Pulmonary Disease.

**Table 5 pharmaceutics-13-01768-t005:** Patient Parameters in Device and Control Groups.

Patient Parameters		Device Group	Control Group	*p* Value
Symptom Distribution *n* (%)	Cough	25 (71.4%)	27 (67.5%)	*p* > 0.05
	Mucus	10 (28. 5%)	1 (2.5%)	*p* < 0.01
	Dyspnea	32 (91.4%)	23(57.5%)	*p* < 0.01
Hypoxemia	Hypoxemic	33 (94.3%)	11 (27.5%)	*p* < 0.01
vs. Room air *n* (%)	Normoxemic	2 (5.7%)	29 (72.5%)	*p* < 0.01
Clinical Parameters	Fewer °C ± SD	36.6 ± 0.4	37.4 ± 0.8	-
	Sp0_2_ (with 0_2_ supplementation)% ± SD	95.0 ± 2.5	93.8 ± 2.89	-
	CRP ^1^ median (mg/L)	41	72	*p* < 0.01
	CRP ^1^ min-max mg/L	1–232	2–372	
	Ferritin median (ng/mL)	698	487	*p* < 0.01
Laboratory Parameters	Ferritin min-max (ng/mL)	102–3713	23–5785	
	Leukocyte median (10^3^/uL)	8400	5675	*p* < 0.01
	Leukocyte min-max (10^3^/uL)	3000–45,200	2250–13,610	
	Neutrophil/Lymphocyte median	11.28	5.22	*p* < 0.01
	Neutrophil/Lymphocyte min-max	1.45–27.66	0.97–20.86	

^1^ CRP: C-Reactive Protein.

**Table 6 pharmaceutics-13-01768-t006:** Oxygen Therapy Method for Device Group and Control Group on the 1st and 10th Days of Treatment.

Patient Classification	Treatment Day 1		Treatment Day 10	
Oxygen Supply Method	Device *n* (%)	Control *n* (%)	Device *n* (%)	Control *n* (%)
0: Room Air	2 (5.7%)	16 (40%)	27 (77.1%)	29 (72.5%)
1: Nasal Cannula	13 (39.5%)	15 (37.5%)	5 (14.3%)	6 (15%)
2: Reservoir Oxygen Mask	12 (31.6%)	7 (17.5%)	2 (5.7%)	1 (2.5%)
3: High Flow Oxygen	8 (23.7%)	2 (5%)	1 (2.9%)	1 (2.5%)
4: Intubation	0 (0%)	0 (0%)	0 (0%)	3 (7.5%)

## Data Availability

Anonymous patient data will be available upon completion of the clinical trials and publication of the completed clinical result study upon request to the corresponding author, under the condition as to not compromise pending patent applications. Proposal requests will be reviewed and approved by the corresponding author, and researchers based on scientific merit and absence of competing interests. The data will include results that are reported in this article, along with other forms if deemed necessary. Upon the approval of the proposal, the data can be transferred through secure online platform following a data access agreement and a confidentiality agreement.

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
