# Peer review of "Early Effects of Low Molecular Weight Heparin Therapy with Soft-Mist Inhaler for COVID-19-Induced Hypoxemia: A Phase IIb Trial"

_pharmaceutics, 2021, doi:10.3390/pharmaceutics13111768_

Round 1
Reviewer 1 Report
The manuscript entitled “Early effects of low molecular weight heparin therapy with soft-mist inhaler for COVID-19-induced hipoxemia: a phase IIb trial”, deals with an interesting topic. The repurposing of heparin for COVID-19 treatment administered as a soft mist by oral inhalation is an innovative proposal and promising results are provided by the study.
Despite the relevance of the work done, the manuscript may be improved. Extended explanation of the methodology should be provided. The relevant points to address are in the attached file.
INTRODUCTION
- Literature on previous work about inhaled heparin might be commented
MATERIALS AND METHODS
- Statitical análisis. the criteria used for sample size are not clear. The paragraph in page
4, lines 120-130, is confusing. Sample size should be based on dispersión of compared
data. The statemet “In this hypothesis, since 85% of patients are discharged at 10 days in
case of direct administration of inhaled LMWH to the lung, it was estimated that the discharge
rate at Day 10 would be 48% for patients who took only standard therapy. On this basis, it was
decided that inclusion of at least 37 patients in each group would provide an adequate sample
size with 90% power and 5% error” is not clear and this does not explain n=40. Besides, n
= 35 for Device Group, due to low compliance
- Patient assignation to Device or Control Groups is not clear. In page 3, line 99-100 it is
said ”patients with relatively worse clinical course were given priority, and classed as the
“Device Group”, but in page 6, lines 180-181 “there were no significant difference in
radiological severity between two groups”.
- Since patients were categorized into 5 “severity” levels, the number of patients into
each level for Control and Device Groups should be considered and provided.
- Pulmonary administration conditions? Dose= 4000 IU is the only information provided.
Volume of inhaled sample? time for inhalation? Manual or mechanic actuación of the
accesory?
- Page 3, lines 103-195: “Patients in the Device Group were given LWMH via soft mist inhaler
at a dose of 4000 IU per administration twice a day, plus the standard treatment”. Does this
mean that Device Group patients received both inhaled and subcutaneous heparin? In
that case, the improvement observed in the Device Group might be due to the higher
dose of heparin in this group, instead of inhaled route. Please clarify, this issue is
relevant and might condition the final conclusion.
- The section “Mechanism of Pulmospray® and in vitro lung deposition” might be
extended to provide more information about this issue. Volume of sample?
Pressure?? Number of replicates??. The same about the results of in vitro study
RESULTS
- Page 8, lines 211, 212 The mean time from onset of symptoms to hospitalization in
the Device Group was 3.5 and 4.4 days in the Control Group” Mean values must be
given together with data dispersión (standard deviation, variation coefficient). Please
prive
- Do the in vitro lung deposition results come from a single experiment?
DISCUSSION
- Page 11, lines 311-313 “This soft-mist inhaler was re-designed by the directives of our
research group to achieve a rate of retention in the lungs, a minimum of twice high than any
other comparable device”. This statement should be supported by literature or
experimental own data. The same regarding the sentence in page 11 , lines 319-320
Please explain
- The last sentence “…. sotf mist LMWH attenuates lung injury and hypoxemia in COVID-19”
may be questioned if the patients from the Device Group received higher dose of
heparin than patients from the Control Group
Author Response
Reviewer 1
1.“Literature on previous work about inhaled heparin might be commented’’
Following the reviewer's wishes, we have now added the following sentence to the manuscript on page 2, lines 65-69;
“Although published studies support only the in vitro effect [16,17] and the conventional usage [16,18] of Unfractioned Heparin (UFH) or LMWH on COVID-19 patients, recent clinical literature focus primarily on the anti-viral efficacy of heparin in COVID-19. Given that inhaled administration of heparin has been proven to be safe in humans, the hypothesis of the research topic may open up unprecedented possibilities [19].”
The paragraph that had the reviewer confused has been revised on page 4, lines 133-142.
To further enlight the reviewer we have included our reasoning in our response given below:
Evaluation of the sample size calculation was carried out depending on the two groups (device group and control group) and the O2 requirement on the 10th day of treatment.
In Accordance with the O2 requirement for both groups; more than one sample size was calculated beginning from 50% (assuming the ranges to be: Type Error 5%, Type II error 0.10%, Power 0.90), and the decision was made to name the largest sample size calculation as the study size group.
- Approximately 14 patients per group if the difference between the two groups is 50% (0.90% vs 0.40%)
- Approximately 20 patients per group if the difference between the two groups is 45%
(0.85% vs 0.40%)
- Approximately 27 patients per group if the difference between the two groups is 40%
(0.80% vs 0.40%)
- Approximately 32 patients per group if the difference between the two groups is 37%
(0.77% vs 0.40%)
- Approximately 37 patients per group if the difference between the two groups is 35% (0.75% vs 0.40%)
For each group, 37 patients were calculated as the minimal sample size when the difference was %35 between the two groups. Depending on this value, the study scope was determined based on 80 patients, 40 for each group. Since more than one possible O2 change could occur between groups, 32 patients would have been sufficient for the sample size of the study.
The answer regarding the reviewers concern was presented on page 10, Table 6.
- ‘’Pulmonary administration conditions? Dose= 4000 IU is the only information provided. Volume of inhaled sample? time for inhalation? Manual or mechanic actuación of the accesory?’’
We agree with the reviewer about the stated issues. Following, we have changed the mentioned sentences to on-page 3 lines 108-111;
“ Patients in the Device Group were given LWMH via a soft mist inhaler at a dose of 4000 IU/0,4 ml per administration twice a day, plus the standard treatment. Each inhaled LMWH application was performed manually for approximately 2 minutes under the supervision of the health professional.’’
- ‘’Page 3, lines 103-195: “Patients in the Device Group were given LWMH via soft mist inhaler at a dose of 4000 IU per administration twice a day, plus the standard treatment”. Does this mean that Device Group patients received both inhaled and subcutaneous heparin? In that case, the improvement observed in the Device Group might be due to the higher
dose of heparin in this group, instead of inhaled route. Please clarify, this issue is
relevant and might condition the final conclusion.’’
Although we agree that this is an important consideration, this was not the case for our study. The standard therapy included subcutaneous heparin as to not deprive the patients to the rights of treatment. This study aims to achieve local targeting of the lungs, a concept that other drug delivery methods are unable to perform. By the time the given dose reaches the lung, only a minimized amount would be given in reality. An increase in dosage would be necessary, which could be fatal in the case of heparin. Both the inhaler device and the drug particle sizes were chosen so that the drug would accumulate in the bronchus and bronchioles of the lungs. In contrast, to achieve a systemic effect smaller aerosol sizes are required that could reach the alveolus. Even if the drug had reached the alveolar region, this amount would not have been enough to present a systemic effect. Moreover, we have also presented our NGI data in the manuscript which further proves that LMWH given with the specific device achieves particules that are able to obtain suitable sizes for local targeting. The necessary information can be found on page 11.
The following was added to the manuscript on page 3 lines 111-115 regarding the concern shown by the reviewer;
“Standart treatment was given to both groups. While the Control Group received systemic treatment by taking only subcutanous LMWH, the Device Group received localy effective LMWH inhalation in addition to the systemic treatment. Administreted inhaled LMWH is expected to accumulate locally and would not increase the systemic dose of LMWH.”
- ‘’The section “Mechanism of Pulmospray® and in vitro lung deposition” might be extended to provide more information about this issue. Volume of sample? Pressure?? Number of replicates??. The same about the results of in vitro study’’
We agree with the suggestion given here and have extended the necessary information both in the methods and results section. Data was already available at the time but it was not included in the manuscript. You can find the extended version on page 7, lines 175-195 and on page 11, lines 302-316.
- ‘’Page 8, lines 211, 212 The mean time from onset of symptoms to hospitalization in the Device Group was 3.5 and 4.4 days in the Control Group” Mean values must be given together with data dispersión (standard deviation, variation coefficient). Please prive.’’
We would like to thank the reviewer for pointing out the missing information that was not previously provided. We have revised the mentioned sentences on page 9 lines 237-239 to “The mean time from onset of symptoms to the hospitalization in the Device Group was 3.50 ±1.99 ( Coefficient of Variation:CV: 56.85%) and 4.40 ±4.23 (Coefficient of Variation: CV: 96,13%) days in the Control Group.’’
- ‘’Do the in vitro lung deposition results come from a single experiment?’’
We like to show your gratitude to the reviewer for pointing out this missing information. The in vitro lung deposition results come from an experiment of n=6. The necessary adjustment has been made on page 7, line 194.
- ‘’Page 11, lines 311-313 “This soft-mist inhaler was re-designed by the directives of our research group to achieve a rate of retention in the lungs, a minimum of twice high than any other comparable device”. This statement should be supported by literature or experimental own data. The same regarding the sentence in page 11 , lines 319-320’’
The given sentence has been revised to include a specific inhaler device and the crucial reference has been added on page 13 lines 349-350.
- ‘’The last sentence “…. sotf mist LMWH attenuates lung injury and hypoxemia in COVID-19” may be questioned if the patients from the Device Group received higher dose of
heparin than patients from the Control Group’’
An important point has been raised here that was also indicated in question 6. We have already explained our reasoning in the previous question and thus have decided not to make any further adjustments.
Once again, we thank you for the time that you put into reviewing our paper and look forward to meeting your expectations. Since your inputs have been precious, in the eventuality of a publication, we would like to acknowledge your contribution explicitly.
The authors’

Reviewer 2 Report
the ide behind your study is fine but there are some major issues which need to be corrected, clarified
- paper formatting: the methods and results are mixed. please describe the methods in the methods section and the results (including tables with patients values which should only be in results section). Please add p value in table 5 . Table 1 inclusion and exclusion criteria are for both device and control group. if not please define the criteria for inclusion in the control group. do you have ethical approval of the study? please briefly describe in the study how informed consent was obtained. Please clearly mention in the methods section which was the duration of LMWH administration
- In vitro experiment: its purpose is not clearly described, and the nature of the device (if it is a pMDI if it is other) is not clear. the results of the in vitro experiment are not clearly discussed, are they important findings for further evaluation of LMWH via inhalation route? if so which would be the next step and why?
- clinical study: you say that the primary outcome was saturation ad hypoxemia status after 10 day therapy with LMWH. This primary outcome is not very appropriately defined but if we consider it as in the present form we cannot find results on it in your draft. Instead something related to the interface of oxygen administration which is an indirect and rather informal indicator of the severity of the gas exchange abnormalities but does not say anything about SaO2/SpO2 dynamics as defined in the primary outcome.
criteria of severity for inclusion in the device group are not clear. the simple consideration is not enough
it is not clear if optimum dose and duration for LMWH could be defined as a result of your study
Author Response
Reviewer 2
- ‘’Paper formatting: the methods and results are mixed. please describe the methods in the methods section and the results (including tables with patients values which should only be in results section).’’
The methods and results section has been updated according to the reviewer’s comment; Table 4 (currently Table 3) has been moved to page 6 along with the given paragraph lines 159-161 and the previous Table 3 is placed in the results section on page 8 as new Table 4.
- ‘’Please add p value in table 5 . Table 1 inclusion and exclusion criteria are for both device and control group. if not please define the criteria for inclusion in the control group. do you have ethical approval of the study? please briefly describe in the study how informed consent was obtained. Please clearly mention in the methods section which was the duration of LMWH administration ‘’
As suggested by the reviewer, we have indicated that inclusion and exclusion criteria are for both device and control groups in Table 1, and the p value was added for Table 5. Our clinical study was approved by the Istanbul Medical Faculty and Turkish Medicines and Medical Devices Agency Clinical Research Ethics Committee (approval number E-66175679-514.03.01-328141, January 27, 2021). Also, this trial is publicly available online at ClinicalTrials.gov with the registration number NCT04990830. Patients and their first-degree relatives were given a briefing concerning the scope and goal of the study prior to enrollment. For those patients who were not able to give informed written consent, approval was obtained from the patient’s first-degree relatives. The duration of LMWH administration has been corrected on the manuscript on Page 3 line 110.
- ‘’ In vitro experiment: its purpose is not clearly described, and the nature of the device (if it is a pMDI if it is other) is not clear. the results of the in vitro experiment are not clearly discussed, are they important findings for further evaluation of LMWH via inhalation route? if so which would be the next step and why? ‘’
As suggested by the reviewer, we have revised the mechanism and in vitro lung deposition studies of Pulmospray® on page 7, lines 175-195. Also, we thoroughly explain the in vitro lung deposition study results and rationalize the importance of the results on page 11, lines 294-305. The importance of these results is that they indicate that most of the LMWH inhalation solution acquired droplets deposits in the bronchus and bronchiole region of the lung which is relevant to our clinical findings. This implies that inhaled dose of LMWH reaches local efficiency and carries out a therapeutic effect on the targeted region.
Soft-mist inhalers are a new classification group that can deliver aerosolized solution. Although previous literature could have classified soft-mist inhaler under pMDI, this is not the case for the device used in the study here.
- ‘’ Clinical study: you say that the primary outcome was saturation ad hypoxemia status after 10 day therapy with LMWH. This primary outcome is not very appropriately defined but if we consider it as in the present form we cannot find results on it in your draft. Instead something related to the interface of oxygen administration which is an indirect and rather informal indicator of the severity of the gas exchange abnormalities but does not say anything about SaO2/SpO2 dynamics as defined in the primary outcome. ‘’
As suggested by the reviewer, we have revised the primary outcome definition as follows in page 5 lines 146-155;
“At the beginning of the trial, patients were categorized into 5 “severity” levels (Table 2) in accordance with the oxygen therapy requirement. Above 95% was the accepted oxygen saturation (SpO2) value to prevent or reverse organ damage and maintain the necessary oxygenation circulation of the tissues in COVID-19, thus it was determined as the primary end point [23]. If the oxygen saturation was higher than 95%, these patients were defined as “Room Air” and this group consisted of patients with milder clinical presentation. At the end of the trial, it was perceived that 13 patients had changed status up to “Room Air” in the Control Group, while for the Device Group the same improvement was observed for 25 patients. The improvement in the severity levels for the device group was significant (p<0.01) compared to the Control Group.”
- 5. ‘’ Criteria of severity for inclusion in the device group are not clear. The simple consideration is not enough it is not clear if optimum dose and duration for LMWH could be defined as a result of your study. ‘’
The severity criteria for the inclusion group was determined by taking into account the oxygen requirement of the patients in order to acquire an oxygen saturation that is higher than 95%. The differences in the severity levels between the device group and control group were given in Table 5. The dose determination of inhaled LMWH was deduced from previously conducted inhalation studies. In similar studies we can observe that up to 10000 IU/day of heparin was given to the patients, and this dose was well-tolerated and no serious adverse effects had been reported (1,2). For our study, the amount of LMWH given is 4000 IU/day. Since this is a phase IIb study, the optimum dose and duration of inhaled LMWH treatment will be determined after a phase III study has been conducted on patients with COVID-19.
References
1- Elsharnouby NM, Eid HE, Abou Elezz NF, Aboelatta YA. Heparin/N-acetylcysteine: an adjuvant in the management of burn inhalation injury: a study of different doses. J Crit Care. 2014 Feb;29(1):182.e1-4. doi: 10.1016/j.jcrc.2013.06.017. Epub 2013 Aug 8. PMID: 23932140.
2- Miller AC, Elamin EM, Suffredini AF. Inhaled anticoagulation regimens for the treatment of smoke inhalation-associated acute lung injury: a systematic review. Crit Care Med. 2014 Feb;42(2):413-9. doi: 10.1097/CCM.0b013e3182a645e5. PMID: 24158173; PMCID: PMC3947059.
Once again, we thank you for the time that you put into reviewing our paper and look forward to meeting your expectations. Since your inputs have been precious, in the eventuality of a publication, we would like to acknowledge your contribution explicitly.
The authors’

Round 2
Reviewer 1 Report
The manuscript has been improved. The sample size calculation is still not clear but I find the present form suitable for publication
Reviewer 2 Report
the authors made efforts to improve the quality of the information in the revised draft